# Relevance of Aquaporins for Gamete Function and Cryopreservation

**DOI:** 10.3390/ani12050573

**Published:** 2022-02-24

**Authors:** Ariadna Delgado-Bermúdez, Jordi Ribas-Maynou, Marc Yeste

**Affiliations:** 1Biotechnology of Animal and Human Reproduction (TechnoSperm), Institute of Food and Agricultural Technology, University of Girona, ES-17003 Girona, Spain; ariadna.delgado@udg.edu (A.D.-B.); jordi.ribasmaynou@udg.edu (J.R.-M.); 2Unit of Cell Biology, Department of Biology, Faculty of Sciences, University of Girona, ES-17003 Girona, Spain

**Keywords:** mammals, oocyte, sperm, water channels, physiology, cryopreservation

## Abstract

**Simple Summary:**

The interaction between cells and the extracellular medium is of great importance; changes in medium composition can drive water movement across plasma membranes. Aquaporins (AQPs) are membrane channels involved in the transport of water and some solutes across membranes. When sperm enter the female reproductive tract after ejaculation, they encounter a drastic change in extracellular composition, which leads to water flowing across the plasma membrane. This triggers a series of events that are crucial to allowing fertilization to take place, such as regulation of sperm motility. In the context of assisted reproduction techniques (ART), long-term storage of gametes is sometimes required, and, during cryopreservation, these cells undergo drastic changes in extracellular medium composition. As a result, AQPs are crucial in both sperm and oocytes during this process. Cryopreservation is of considerable importance for fertility preservation in livestock, endangered species and for individuals undergoing certain medical treatments that compromise their fertility. Further research to fully elucidate the roles and underlying mechanisms of AQPs in mammalian sperm is therefore warranted.

**Abstract:**

The interaction between cells and the extracellular medium is of great importance, and drastic changes in extracellular solute concentrations drive water movement across the plasma membrane. Aquaporins (AQPs) are a family of transmembrane channels that allow the transport of water and small solutes across cell membranes. Different members of this family have been identified in gametes. In sperm, they are relevant to osmoadaptation after entering the female reproductive tract, which is crucial for sperm motility activation and capacitation and, thus, for their fertilizing ability. In addition, they are relevant during the cryopreservation process, since some members of this family are also permeable to glycerol, one of the most frequently used cryoprotective agents in livestock. Regarding oocytes, AQPs are very important in their maturation but also during cryopreservation. Further research to define the exact sets of AQPs that are present in oocytes from different species is needed, since the available literature envisages certain AQPs and their roles but does not provide complete information on the whole set of AQPs. This is of considerable importance because, in sperm, specific AQPs are known to compensate the role of non-functional members.

## 1. Introduction

While the composition of biological membranes is responsible for their intrinsic, but highly limited, water permeability, the transport of water and metabolites through the plasma membrane is of high importance for the maintenance of cell function and survival [1]. Mechanisms other than simple diffusion are thus required to allow water and solutes to pass across the plasma membrane for several cell functions. In 1988, Agre and his colleagues identified a particular 28 kDa membrane protein in erythrocytes, which was different from Rh blood group membrane proteins [2]. This protein was characterized in 1991 as a transmembrane channel [3] and was named as CHIP28. In 1992, the protein was confirmed to be involved in water transport through its exogenous expression in *Xenopus laevis* oocytes. That overexpression increased the permeability to water, and the inhibition of CHIP28 with mercury chloride, a well-known inhibitor of water channels, was observed to reduce water permeability [4], and the channel was later renamed aquaporin 1 (AQP1).

The aim of the present review is to discuss the function of aquaporins (AQPs) in mammalian gametes. For this purpose, this work includes a classification and description of the structure and function of AQPs. The presence of different AQPs in mammalian sperm and oocytes is thus described and their roles are discussed, including the involvement of AQPs in gamete cryopreservation. The article also refers to the different strategies that have been used to explore the physiological characteristics of AQPs, including knockout models, exogenous expression and inhibition through pharmaceutical agents.

## 2. Water Channels in the Plasma Membrane of Mammalian Cells: The Family of Aquaporins

Aquaporins (AQPs) are a family of transmembrane proteins that are present ubiquitously in all species and cell types whose main function is to allow the transport of water across cell membranes (reviewed in [5]). In addition, some members of this family also exhibit permeability to small molecules, such as glycerol [6], urea [7], ammonia [8], hydrogen peroxide [9] and arsenite [10].

The members of the AQP family present a highly conserved gene sequence, structure and function. In terms of structure, AQPs present six transmembrane α-helices (TM1-6; Figure 1A) surrounding a central pore that, when resolved through X-ray, shows an “hourglass” shape (Figure 1B) [11,12]. Transmembrane segments are connected by loops (A–E); loops B and E each present an NPA motif (asparagine, proline, alanine), which is the typical AQP signature motif (reviewed in [13]). The two loops that present an NPA motif are half-transmembrane helices that locate towards the center of the channel and create a broken seventh-transmembrane helix (Figure 1A,B; reviewed in [5]). Water molecules cross AQPs via a single-molecule well and interact with the lateral chains of the amino acids that form the channel through hydrogen bonds. In this sense, the NPA motif disrupts any potential proton conduction locking the central molecule of water in a conformation that avoids molecular reorientation [14]. Another highly preserved feature in AQP structure is the selectivity filter, which is the narrowest point in the channel and is considered to be essential for channel selectivity [14]. The region near these residues is also known as aromatic/arginine (ar/R) constriction region (Figure 1A), since the arginine is a highly conserved residue among all AQPs [15].

Aquaporins (AQPs) function as tetrameric structures formed by four monomers, each with their own permeable pore (Figure 1C). Tetramers seem to be stabilized through the interaction of TM1 and TM2 of one monomer with TM4 and TM5 of the adjacent one. While an extra pore is known to be formed in the center of the tetramer, its actual role remains, to the best of our knowledge, largely unknown (reviewed in [16]). Moreover, tetrameric structures appear to be crucial for AQPs’ mechanosensitivity (Figure 2), as it has been described for AQP1 and AQP4 [17,18]. Increasing tension in the plasma membrane, which indicates progressive cell swelling, causes a decrease in the water transport rate of AQPs that evidences the closing of the water pore. This might be driven by slight distortions in the monomers in response to membrane stretching, which would alter the formation of hydrogen bonds between AQP amino acids and water molecules inside the pore, leading to a decrease in water permeability (reviewed in [16]).

In spite of having a highly preserved sequence and structure, the separate members of the AQP family differ in certain structural features and specific permeability to water and other molecules. AQPs can, therefore, be classified into three different groups based on variations in their structure and permeability: orthodox AQPs, aquaglyceroporins (GLPs) and superAQPs (reviewed in [19]; (Figure 3).

**Figure 2 animals-12-00573-f002:**
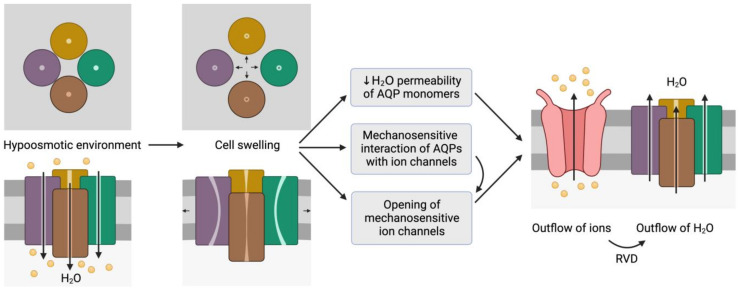
Representation of the mechanosensitive mechanism of aquaporins (AQPs). When cells are in a hypoosmotic environment, water enters them in response to the concentration gradient of solutes. Due to the entry of water, cells may undergo swelling, which can cause the distortion of AQP structures and thus compromise water permeability. In addition, AQPs may have mechanosensitive interactions with ion channels that can trigger their opening to trigger the outflow of ions. In fact, some ion channels are mechanosensitive and open in response to cell swelling, regardless of AQP signaling. The outflow of ions generates a driving force that elicits the outflow of water; this process is also known as regulatory volume decrease (RVD). This schematic representation is inspired by the model proposed by Hill et al. [20] and its representation by Ozu et al. [16].

### 2.1. Orthodox Aquaporins (AQPs)

The group of orthodox AQPs includes AQP0, AQP1, AQP2, AQP4, AQP5, AQP6 and AQP8. Orthodox AQPs present the smallest channel size among the different groups of AQPs and their selectivity filter has a highly hydrophilic nature [14]. These two characteristics are the reason for the exclusive permeability of this group of AQPs to water through a steric mechanism of selectivity (reviewed in [24]). In addition, the NPA motif also plays a function in selectivity because, in orthodox AQPs, the side chains of the amino acids of this motif narrow the pore to a smaller diameter and are more hydrophilic than those of the other family members [14]. The only exception in this group is AQP6, which also presents anion permeability at low pH [25] and localizes in the membrane of intracellular organelles; remarkably, this differs from the other orthodox AQPs that localize in the plasma membrane [26].

### 2.2. Aquaglyceroporins (GLPs)

The group of GLPs includes AQP3, AQP7, AQP9 and AQP10. Water molecules cross GLPs through the establishment of the same interactions as with orthodox AQPs and also form a single-molecule well. In GLPs, nevertheless, the diameter of the pore, which is limited by the NPA motif, is larger than that of orthodox AQPs. In addition, the selectivity filter has a hydrophobic patch of residues and the region near the NPA motif is less hydrophilic than in orthodox AQPs, which determines the lower hydrophilicity of this group of AQPs [14]. These two characteristics are crucial to understand why bigger molecules are, together with water, able to pass through GLPs.

It is worth noting that glycerol is transported with preference to water by this group of AQPs. In mammals, glycerol permeation is known to be important during fasting states, when it is obtained from the degradation of triglycerides in adipocytes. It has been suggested that in hepatocytes where gluconeogenesis occurs, AQP9 and AQP7 could be the entry and exit routes of glycerol, respectively (reviewed in [27]). Furthermore, glycerol is used as a permeable cryoprotectant for cell preservation [28], which is possible because the presence of GLPs in the plasma membrane allows its movement.

Arsenite can also permeate AQP7 and AQP9. As leukocytes express AQP9, arsenite has been suggested to have a potential pharmacological use in the treatment of promyelocytic leukemia [10]. In fact, AQP9 shows a promiscuous permeability to a wide range of non-charged solutes, including polyols, purines (adenine), pyrimidines and urea, but is impermeable to cyclic sugars, nucleosides, amino acids and charged ions [29].

### 2.3. Superaquaporins (superAQPs)

The last group, also known as superAQPs, includes AQP11 and AQP12, which are expressed in intracellular membranes [30,31]. In fact, they present an endoplasmic reticulum (ER) retention signal at the carboxy-terminal region (reviewed in [32]), and AQP11 has been identified in the ER when exogenously expressed in mammalian cells [30]. Moreover, these two AQPs present lower homology to the other groups compared with the identity that can be observed between orthodox AQPs and GLPs (reviewed in [33]). On the one hand, the members of this group of AQPs present a variation in the sequence of the NPA motif of loop B, whereas it is preserved at loop E. In loop B of superAQPs, alanine is replaced by cysteine; AQP11 and AQP12, therefore, have a NPC motif instead (reviewed in [32]). On the other hand, there are highly conserved sequences before the first NPA motif and after the second one that are also different in superAQPs compared to the other members of the family of AQPs (reviewed in [32]). As far as permeability is concerned, while superAQPs are involved in the transport of water, they have also been identified as glycerol channels in human adipocytes [34].

## 3. Aquaporins in Mammalian Sperm

The presence and localization of AQPs in male gametes differ between species and cell domains. A summary of the members of this family that have hitherto been explored in gametes is shown in Table 1. (This Table includes the different species in which they have been studied in a chronological order.) In sperm, the presence of AQPs has been assessed through Western blotting, whereas their localization has been assessed by means of immunocytochemistry/immunofluorescence. To date, AQP3, AQP7, AQP8 and AQP11 are the most investigated proteins in mammalian sperm, although AQP1 and AQP9 have also been identified in certain species (Table 1, Figure 4).

Aquaporin 3 (AQP3) has been identified in pig [35], cattle [36,37], horse, mouse [38] and human [39,40] sperm [41]. Immunolocalization experiments have determined its presence in the sperm mid-piece in pig and cattle, whereas in human and mouse sperm it has been identified in the principal piece. Aquaporin 7 (AQP7) has been identified in pig [42,43], cattle [36,37], horse, human [39,44,45,46], mouse [47] and rat [48,49,50] sperm [41]. It is present in the tail of ejaculated sperm, and has also been identified in certain regions of the head of human sperm [39,46]. Aquaporin 8 (AQP8) is present in human [39,45], mouse and rat sperm [47] and has been found to be located in the sperm tail. In humans, AQP8 has been suggested to localize in mitochondria from the mid-piece [39]. Aquaporin 11 (AQP11) is present in pig [43], horse [41], mouse [51], rat [51] and human sperm [39] and is mainly localized in intracellular structures from the sperm tail. In humans, nevertheless, it has also been identified in the sperm head [39] and, in rats, it is exclusively present in the terminal piece of the sperm tail [51].

On the other hand, while different studies have failed to identify AQP1 in mammalian sperm cells, including those from sheep [52], humans [45,53] and mice [54], this protein was purported to be expressed in dog sperm [55]. Aquaporin 9 (AQP9) has been identified in the head of pig sperm [42] but not in that of human [45] or mouse sperm [47].

**Table 1 animals-12-00573-t001:** Members of the family of aquaporins (AQPs) identified in gametes from different species of mammals, including ram, pig, horse, cattle, dog, mouse, rat and human. The different species in which they have been studied are presented in a chronological order.

Aquaporin	Gamete	Species	Strategy of Detection	Methodology	References
AQP1	Sperm	Not in ram	Protein	WB	[52]
		Not in human	mRNA, protein	WB, RT-PCR	[45,53]
		Not in mouse	Protein	IHC	[54]
		Dog	mRNA, protein	RT-PCR, WB	[55]
	Oocyte	Not in mouse	mRNA	RT-PCR	[56]
		Not in rat	mRNA	RT-PCR	[57]
		Human	mRNA, protein	sc-RNAseq, IF	[58]
AQP2	Oocyte	Not in mouse	mRNA	RT-PCR	[56]
		Not in rat	mRNA	RT-PCR	[57]
		Human	mRNA, protein	sc-RNAseq, IF	[58]
AQP3	Sperm	Mouse	Protein	IF, IGEM	[38]
		Human	mRNA, protein	IF, WB, ICC, RT-PCR, FC	[38,39,40]
		Pig	Protein	WB, IF	[35]
		Horse	Protein	WB	[41]
		Cattle	Protein	WB, ICC	[36,37]
AQP3	Oocyte	Not in rat	mRNA	RT-PCR	[57]
		Mouse	mRNA, protein	RT-PCR, IF	[56,59]
		Human	mRNA, protein	RT-PCR, IF	[59]
		Pig	mRNA	RT-PCR	[60]
		Cattle	mRNA	RT-PCR	[61]
AQP4	Oocyte	Not in mouse or rat	mRNA	RT-PCR	[56,57]
AQP5	Oocyte	Not in mouse or rat	mRNA	RT-PCR	[56,57]
AQP6	Oocyte	Not in mouse or rat	mRNA	RT-PCR	[56,57]
AQP7	Sperm	Rat	Protein	IF, WB	[49,50]
		Human	mRNA, protein	WB, ICC, IF, RT-PCR, FC	[39,44,45,46]
		Mouse	mRNA, protein	WB, RT-PCR	[47]
		Pig	Protein	WB, ICC	[42,43]
		Cattle	Protein	WB, ICC	[36,37]
		Horse	Protein	WB	[41]
	Oocyte	Mouse	mRNA	RT-PCR, IF	[56,59]
		Not in rat	mRNA	RT-PCR	[57]
		Human	mRNA, protein	RT-PCR, IF	[59,62]
AQP8	Sperm	Mouse	mRNA, protein	WB, RT-PCR	[47]
		Human		WB, ICC, RT-PCR, FC	[39,45]
	Oocyte	Not in mouse or rat	mRNA	RT-PCR	[56,57]
AQP9	Sperm	Not in mouse	Protein	WB	[47]
		Not in human	mRNA, protein	WB, ICC, RT-PCR, FC	[45]
		Pig	Protein	WB, ICC	[42]
	Oocyte	Not in mouse	mRNA	RT-PCR	[56]
		Rat	mRNA	RT-PCR	[57]
		Human	Protein	IF	[62]
AQP11	Sperm	Mouse	Protein	IHC	[51]
		Rat		IHC	[51]
		Pig		WB, ICC	[43]
		Human		WB, ICC	[39]
		Horse		WB	[41]
		Cattle		IHC, WB	[63]
	Oocyte	Human	mRNA, protein	sc-RNAseq, IF	[58]

Abbreviations: WB, Western blotting; RT-PCR, real-time PCR; IHC, immunohistochemistry; sc-RNAseq, single cell RNA sequencing; IF, immunofluorescence; IGEM, immunogold electron microscopy; ICC, immunocytochemistry; FC, flow cytometry.

**Figure 4 animals-12-00573-f004:**
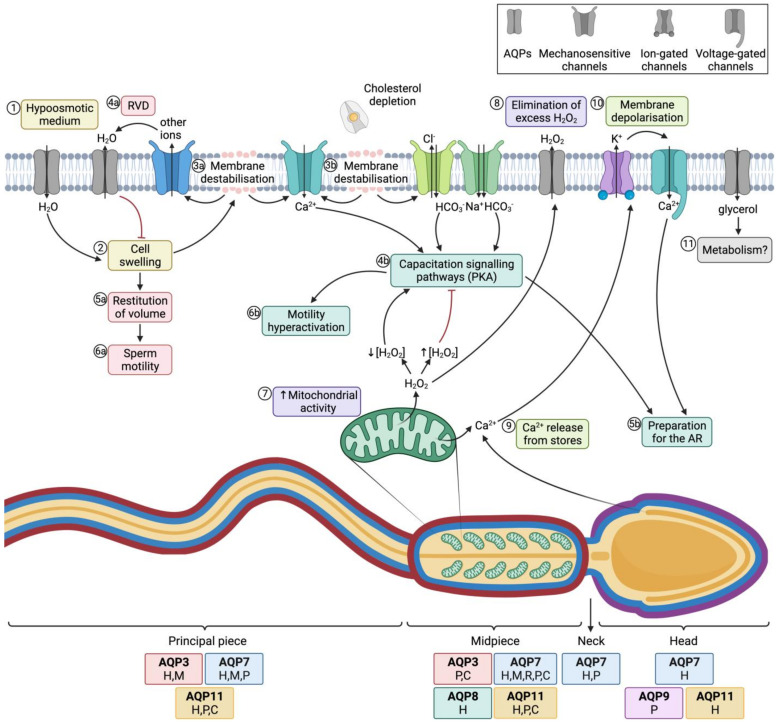
Aquaporins (AQPs) and sperm function after ejaculation. (1) After entering the female tract, sperm encounter a hypoosmotic medium, which causes water influx and thus (2) cell swelling. (3a) The increase of cell volume causes membrane destabilization (which affects the sperm plasma membrane, but also alters mitochondrial and acrosomal membranes). (4a) As a consequence, mechanosensory ion channels are open and outflowing ion currents activate regulatory volume decrease (RVD) events, such as (4a) water outflow. (5a) Cell volume is then restituted, which allows (6a) normal sperm motility. (3b) In the female reproductive tract, membrane destabilization occurs in conjunction with cholesterol depletion. The resulting membrane reorganization elicits the entrance of calcium and bicarbonate through different ion channels, which activates capacitation signaling pathways (4b) that involve the activation of protein kinase A (PKA). (5b) Downstream events prepare the spermatozoon for the acrosome reaction (AR) and (6b) drive hyperactivated motility. (7) In this context, mitochondrial activity is elevated and reactive oxygen species (ROS), such as H_2_O_2_, are produced. This molecule at low concentrations is essential to elicit capacitation, but at high concentrations it acts as an inhibitor of this process. (8) Aquaporins (AQPs) play an essential role, allowing the outflow of the excess amounts of H_2_O_2_ towards the extracellular medium. (9) Another downstream event of the capacitation signaling pathway consists of the release of Ca^2+^ from intracellular stores, which triggers the opening of calcium-activated K^+^ channels that causes, in turn, plasma membrane hyperpolarization (10); membrane hyperpolarization subsequently opens voltage-gated Ca^2+^ channels. This increase in intracellular Ca^2+^ is crucial for acrosome reaction [64,65] (11) Glycerol transport through GLPs can be used by metabolic pathways. The sperm regions where AQPs have been identified in different mammalian species are highlighted in the diagram. Boxes indicate the species in which each AQP has been identified (C, cattle; P, pig; H, human; M, mouse; R, rat).

### 3.1. Sperm Aquaporins (AQPs) and Osmoregulation

The most important role of AQPs in sperm is strictly related to osmoregulation. When sperm enter the female reproductive tract after ejaculation, they undergo a severe osmotic stress, since osmolality in the cauda epididymis is higher than in the female reproductive tract (reviewed in [66]). In fact, during their transit through the epididymis, sperm acquire their osmoadaptability because the extracellular medium is progressively more hyperosmotic from the caput to the cauda. For this purpose, sperm uptake osmolytes from the epidydimal plasma, which allows the counteraction of the lower osmolality in the female reproductive tract after ejaculation (reviewed in [67]). One must also consider that, at the time of ejaculation, sperm encounter the seminal plasma. There are differences between separate species of mammals in terms of differential osmolality between cauda epididymis, seminal plasma and the oviduct. On the one hand, in bovines, the epididymis, uterine and oviductal environments are hyperosmotic compared to the isoosmotic seminal plasma. In humans, mice, and rats, osmolality is progressively lower from the epididymis to the seminal plasma and then to the uterus, whereas in cattle and sheep the uterus and the epididymis are relatively hyperosmotic to seminal plasma (reviewed in [68]).

Hypoosmotic shock causes sperm swelling due to an excessive uptake of water, which ends up impairing the normal movement of the sperm tail, which becomes coiled. In addition, plasma membrane function is highly compromised and if the critical volume is reached the membranes may experience ruptures [69]. Under physiological conditions, hypoosmotic stress initiates the signaling pathway involved in RVD, which activates an osmolyte efflux that drives water movement and restores cell volume [67]. Aquaporins (AQPs) are therefore crucial to allow the rapid trafficking of water across the plasma membrane, which, in turn, regulates cell volume (Figure 4; 1–4a).

Some AQPs, nevertheless, do not seem to be inert pores at the plasma membrane, and, in fact, different mechanisms of regulation of water permeability have been described. Sperm from *Aqp3* knockout mice exhibit an impaired capacity to regulate their volume after entering the uterus, and subsequently show progressive cell swelling [38]. For this reason, Chen et al. [70] hypothesized that if AQP3 function was restricted to facilitate the time required for sperm to reach the osmotic equilibrium, both influx and efflux of water should be diminished and progressive cell swelling would not be observed. As previously stated, water permeability of AQP tetramers decreases in response to plasma membrane tension, probably because of pore channel distortion (reviewed in [16]). Chen et al. [70] thus proposed that AQP3 in sperm could have an additional mechanosensory role that would trigger RVD events via interaction with other ion channels (Figure 2). Remarkably, this involvement of AQPs in the upstream events of RVD has been previously described in salivary gland cells, where AQP5 is coordinated with TRPV4 [71], and in astrocytes, where an AQP4/TRPV4 complex is involved in volume regulation [72]. In this sense, it has been hypothesized that membrane tension would increase the distance between monomers and that these changes in the tetrameric conformation would activate the RVD signaling pathway; in this scenario, interactions with ion channels would evoke the driving force leading to water efflux [70]. 

In sperm from an *Aqp7* knockout (KO) mouse model, water transport was faster than in those from wild-type animals when exposed to quinine, an agent that induces cell swelling through hindering osmolyte efflux and, as a consequence, RVD events [47]. In addition, morphological alterations indicative of cell swelling, such as tail coiling, have not been observed in sperm from *Aqp7* knockout mice [73]. The higher water transport and the absence of cell swelling observed in the KO model could be explained through the compensation of function by other AQPs because higher levels of *Aqp8* mRNA in the testis of *Aqp7*-knockout mice were observed [47]. Taking all these findings together, while AQP7 does not seem to play a relevant role in the osmoregulation of mouse sperm, alterations in the distribution of AQP7 in human sperm have been reported to be negatively correlated to the presence of sperm with coiled tails and other morphological alterations [46].

Aquaporin 8 (AQP8) is also relevant in osmoregulation, as its levels in human sperm have been found to be inversely correlated with the presence of sperm with coiled tails, which is indicative of osmotic stress [45]. The importance of AQP8 in osmoregulation has also been explored through its inhibition through HgCl_2_, which blocks quinine-induced swelling in both human [45] and mouse sperm [47,74]. In addition, Yeung et al. described the importance of AQP8 in water outflow associated with RVD, as its inhibition in mouse sperm through HgCl_2_ reduces cell shrinkage in response to osmolyte currents [47]. It is, nevertheless, worth noting that mercury is an inhibitor of all AQPs except AQP7—since it binds covalently to a cysteine residue that is conserved in all AQPs except AQP7—which blocks water permeability [48,75]. Therefore, even though some studies using mercury as an inhibitor suggested that the observed effects were a consequence of AQP8 blocking, water permeability is also impaired when other AQPs are inhibited.

### 3.2. Aquaporins, Sperm Functionality and Male Fertility

The relationship between AQPs and sperm function has been evaluated in previous studies. Essentially, sperm’s capacity to regulate volume is crucial for fertility, since cell swelling in response to osmotic stress causes tail bending, which can, in turn, affect sperm motility (reviewed in [76]). Nevertheless, osmotic changes in response to the hypotonic shock that sperm undergo when they enter the female reproductive tract are essential to elicit sperm capacitation. On the one hand, changes in sperm volume in response to osmotic shock trigger the opening of mechanosensitive calcium channels and thus elicit calcium influx, which is one of the first events that occurs during capacitation [77]. On the other hand, acrosomal swelling is required for a physiological acrosome reaction [78]. In addition, strict regulation of H_2_O_2_ concentration is essential for motility hyperactivation and acrosome reaction during capacitation, despite the mechanism remaining unknown [79]. Since some members of this family are peroxiporins, their potential involvement in acrosome reaction can be suggested. In addition, the permeability of GLPs to glycerol has been proposed to be relevant for the use of this molecule in metabolic pathways and as a source of energy in sperm [45]. The role of AQPs in sperm fertility is, therefore, evidenced by their involvement in both motility and capacitation-associated events, which are crucial for sperm to successfully fertilize the oocyte (Figure 4).

#### 3.2.1. Aquaporins in Sperm Physiology from Livestock

In pig sperm, AQP11 but not AQP7 levels are positively correlated with sperm motility and different quality and functionality parameters, such as membrane lipid disorder or acrosome integrity, in fresh semen [43]. Delgado-Bermúdez et al. [80] assessed the effects of AQP inhibition during capacitation in pig sperm. The specific inhibition of AQP3 through CuSO_4_ [81] caused an increase in H_2_O_2_ levels, which is in agreement with the function of this protein in cell detoxification. Inhibition of all AQPs except AQP7 through HgCl_2_ impaired sperm motility and induced alterations in plasma and acrosome membrane integrity, which could be a direct consequence of the limited osmoadaptation ability caused by AQP blocking through mercury. Finally, while the inhibition of all AQPs with AgSDZ [82] was found to exert similar effects to those caused by the addition of HgCl_2_, it also led to a decrease in intracellular pH and an increase in tyrosine phosphorylation levels after the induction of the acrosome reaction. This suggests that AQP7 might be able to compensate partially the lack of function of the other AQPs in the presence of mercury, but its blockade through silver avoids this functional compensation [80]. To sum up, in pigs, AQP3 and AQP11 seem to be involved in the regulation of sperm motility, whereas AQP7 might partially compensate the impaired osmoregulation ability when mercury-sensitive AQPs are inhibited, since the acrosome reaction remains unaltered when this protein is functional.

In goats, a recent study investigated the detrimental effects of the presence of mercury on sperm structure [83]. Essentially, mercury was found to cause alterations in plasma, acrosome and mitochondrial membranes, as well as alterations in the axoneme. Together with the results of a previous work [84], in which mercury was added to capacitation medium, the effects of this molecule are in agreement with those described in other species. Remarkably, Kushawaha et al. [83] proposed the potential mechanisms leading to the observed consequences in sperm structure and physiology, but AQPs were not contemplated. Considering the sperm damage observed due to the presence of mercury in different species, a potential role of AQPs in the detrimental impact induced by this agent should not be discarded (Figure 5). 

#### 3.2.2. Aquaporins in Murine Sperm Physiology 

In the study of Chen et al. [38], while sperm motility in an *Aqp3* knockout mouse model seemed to be normal shortly after entry to the female reproductive tract, progressive cell swelling was found to end up causing tail coiling, altering sperm motility. This disrupted motility was observed to compromise the sperm’s ability to migrate into the oviduct and to decrease fertilization, thus reducing male fertility [38]. These results also provide evidence that the impairment of sperm motility in *Aqp3* knockout mice is a consequence of inefficient osmoregulation but not of structural alterations caused by the absence of AQP3 during spermatogenesis. In spite of this, other research found that an *Aqp7* knockout mouse model did not present alterations in sperm motility or production, nor in in vivo and in vitro fertility outcomes [73]. All these data suggest that, at least in mice, AQP3, but not AQP7, seems to have a relevant role in sperm motility.

The potential role of glycerol as an energy source was explored in rat sperm by Cooper et al. [85], who reported the presence of this molecule in rat epididymis and also found it to be involved in sperm metabolism. While, in this scenario, AQP7 could allow the entry of glycerol into sperm, as in other tissues [86], further research is needed to elucidate the relationship between sperm AQP7 and the utilization of glycerol to produce energy.

#### 3.2.3. Aquaporins in Human Sperm Physiology

In human sperm, AQP7 levels are higher in fertile donors than in infertile patients and are correlated with progressive motility [44,45]; however, these differences are not observed in the case of AQP8 levels [45]. Moreover, alterations in the localization pattern of AQP7 in human sperm are negatively correlated with sperm motility and fertility [46]. Such an alteration in motility could be explained by the cell swelling that results from the lower osmoregulatory ability when AQP7 levels are lower. Nevertheless, Yeung [87] suggested that lower intake of glycerol, which has been previously proposed as an energy source in rat sperm (see Section 3.2.1), through AQP7 could contribute to this sperm motility impairment. On the other hand, blocking of human sperm AQPs (except for AQP7) through HgCl_2_ has been reported to cause a drastic decrease in sperm motility, which is reversed through 2-mercaptoethanol, an agent that has a higher affinity than mercury for the -SH group of cysteine residues in AQPs [40]. This indicates that all AQPs are involved in human sperm motility.

In human sperm, when mercurial compounds are used to inhibit AQPs, mitochondrial membrane potential is also altered; hence, structural alterations caused by cell swelling do not seem to be the exclusive cause of motility disruption when AQP function is blocked [40], which is consistent with the mechanisms described in Figure 5. Since osmotic stress causes an increase in ROS and AQP3 and AQP8 are peroxiporins [39,88], the blockade of these channels with mercury could also hinder H_2_O_2_ efflux. In fact, Laforenza et al. [39] evidenced that H_2_O_2_ efflux through AQPs is inhibited by mercury. The intracellular accumulation of ROS due to the lack of detoxification may, in turn, damage cellular organelles, such as mitochondria, which would explain the disruption in MMP and would be directly related to impaired sperm motility. The role of AQPs in sperm physiology, therefore, seems to be common in livestock and humans.

In addition, mRNA and protein levels of AQP3 were reported to be lower in asthenozoospermic patients than in fertile controls [89], which confirms the relationship between this protein and sperm motility. As in asthenozoospermic patients, AQP3 content was negatively correlated with activated caspase-3 (CASP3) levels; Mohammadi et al. [89] suggested that CASP3 could degrade AQP3 and thus alter its function. Since, nevertheless, AQP3 also mediates H_2_O_2_ transport [88], it is reasonable to suggest that the correlation between this apoptosis marker and AQP3 levels might be the consequence of the same sequence of events occurring in the previously described mitochondrial damage. The proposed sequence of events would be that sperm with lower levels of AQP3 have a worse osmoadaptation ability, thus causing oxidative stress and ROS generation. These sperm with lower levels of AQP3 show an impaired ability to eliminate H_2_O_2_, which would accumulate intracellularly and disrupt mitochondrial function through the disruption of MMP, thus leading to an increase in CASP3 levels. In fact, Laforenza et al. [39] unveiled a reduced permeability to water and H_2_O_2_ in subfertile patients, which would underpin the relationship between AQP levels and fertility in human sperm. 

Pellavio et al. [90] recently described the potential involvement of AQPs in the subfertility caused by human papillomavirus (HPV) infection. Immunolocalization and co-immunoprecipitation experiments demonstrated HPV binding to AQP8, but not to AQP3 or AQP7. The HPV, therefore, effectively decreases AQP8 permeability, thus making cells more vulnerable to osmotic and oxidative stresses, which could be the cause of the drastic decrease in sperm motility in the presence of this infection [90].

To sum up, all AQPs investigated thus far in human sperm have been reported to play a relevant role in sperm motility (both AQP7 and other mercury-sensitive proteins, including AQP3), whereas AQP3 and AQP8 look to be involved in ROS elimination.

### 3.3. Aquaporins and Sperm Cryopreservation

Cryopreservation of mammalian sperm is the most efficient method for long-term storage. This procedure is of high importance in livestock production, since cryopreserved sperm can be used for artificial insemination (AI) (reviewed in [91]), but they are also important for the establishment of genetic banks for endangered domestic and wild species [92]. Furthermore, it is also relevant as a procedure for the management of male fertility before undergoing certain treatments or for storing sperm that are to be used later in assisted reproduction technology (ART) [93]. Cryopreservation, however, is a challenging process for sperm integrity because of the osmotic shock these cells endure; under these circumstances, allowing water outflow of the cell is crucial to avoid the formation of intracellular crystal structures. This procedure may affect the integrity of the sperm nucleus, cytoskeleton and plasma membrane, reduce mitochondrial activity and motility and impair protein function [94,95,96], which results in decreased fertilizing ability. To minimize cryoinjuries, freezing media are supplemented with cryoprotecting agents, among which glycerol is used in different livestock species [28]. It is worth considering, nevertheless, that, apart from variations in size and morphology, the marked differences in plasma membrane composition and metabolism between sperm from separate species reflect the high variability in their cryotolerance (also known as freezability). In addition, within a given species, samples with similar fresh semen quality present both intra- and inter-individual differences in terms of sperm cryotolerance. Ejaculates can thus be classified into good (GFE) and poor freezability ejaculates (PFE) depending on post-thawing sperm quality [97,98,99]. In this context, the need for cryotolerance biomarkers has become highly apparent. Recent studies have evidenced differences in the transcriptomes [100], metabolomes [101], antioxidant activity [102,103], lipidomes [104] and proteomes [105,106,107] between GFEs and PFEs.

Among the proteins that have considerable relevance as potential cryotolerance biomarkers, levels of AQPs in sperm from different mammalian species have been found to differ between GFEs and PFEs. In bovine sperm, AQP3 content is positively related to motility after thawing [37], whereas AQP11 levels are correlated with both cryotolerance and AI outcomes of cryopreserved sperm [63]. Also in this species, relative levels of AQP7 in sperm are positively correlated with sperm cryotolerance [36], and those of *AQP7* mRNA are positively correlated with both their osmoregulation ability and fertility [108].

In both pig and horse sperm, AQP3 and AQP7, but not AQP11, are related to sperm cryotolerance; in pig sperm, AQP7, but not AQP9, relocalizes after freeze-thawing [41,42,109]. Moreover, the inhibition of different groups of AQPs in pig [110,111] and horse sperm [112] confirms the involvement of GLPs in sperm cryotolerance and suggests that the relevance of this group of AQPs is higher in GFEs than in PFEs. In these studies, 1,3-propanediol is used as a strong inhibitor of orthodox AQPs [113,114] and a mild inhibitor of GLPs [114]; acetazolamide is used as an inhibitor of orthodox AQPs [115,116], whereas phloretin is used to block GLPs [29,117] These investigations also suggest that not only are GLPs relevant for water permeability and osmoregulation but they are also involved in the transport of glycerol and hydrogen peroxide through the plasma membrane. Related to this, it is worth bearing in mind that, on the one hand, glycerol is a commonly used permeable cryoprotectant for livestock sperm cryopreservation. A quick inflow of this agent through the plasma membrane is, therefore, important to ensure that a sufficient concentration is present intracellularly during freezing and that a rapid outflow of the cell to avoid its toxic effects takes place after thawing. On the other hand, intracellular levels of ROS increase during cryopreservation as a result of mitochondrial dysfunction [118], and blocking of AQP3, which is permeable to H_2_O_2_, results in an intracellular accumulation of H_2_O_2_, which supports its important role in detoxification during freezing and thawing.

## 4. Aquaporins in Mammalian Oocytes

While AQPs in sperm have been mainly studied from a proteomic point of view, their presence in oocytes has been studied through the evaluation of mRNA expression levels. This is due to the fact that obtaining enough protein content from oocytes is more difficult than in sperm, and RT-PCR for the detection of mRNA has a higher sensibility than Western blotting (Table 1). In oocytes, while AQP3 and AQP7 have been found to be expressed in most of the species that have been hitherto investigated, AQP9 has also been identified in some mammals. 

In pig [60] and cattle oocytes [61], only the expression of *AQP3* mRNA has been explored and confirmed to date. Human oocytes have been reported to express *AQP3* and *AQP7* mRNAs [59], although not only the presence of AQP7 protein but also that of AQP9 in germinal vesicle- and MII-oocytes has been confirmed [62]. In addition, Zhang et al. detected the expression of *AQP1*, *AQP2* and *AQP11* mRNAs and their corresponding proteins in human oocytes [58]. Finally, in rat and mouse oocytes, the mRNA expression of *Aqp1–Aqp9* has been evaluated. In mouse oocytes, *Aqp3* and *Aqp7* mRNAs were found [56,59], whereas only in rats was the expression of *Aqp9* mRNA detected, restricted to proestrus oocytes [57]. 

### 4.1. Oocyte Aquaporins and Osmoregulation

It has been put forward that the main function of AQPs in sperm is osmoadaptation and the key events in which these proteins are crucial are the entry to the female reproductive tract and during the cryopreservation process. The function of AQPs in oocytes, therefore, seems to be less crucial in their physiological environment because these cells do not undergo major osmolality changes after ovulation unless they are cryopreserved. For this reason, most of the studies aiming to evaluate the characteristics and functional role of AQPs in oocytes are based on the evaluation of their permeability to water and cryoprotectants.

In pig oocytes, water and cryoprotectants cross the plasma membrane mainly through simple diffusion [60]. However, the expression of exogenous (human and zebrafish) AQP3 also increases their permeability to water and cryoprotectants [119]. 

In bovine oocytes, AQP3 is permeable to water, glycerol and ethylene glycol, but not to DMSO or propylene glycol. In this species, water molecules and cryoprotectants cross the plasma membrane predominantly by simple diffusion, but channel-facilitated transport seems to have a more relevant role in bovine oocytes compared to other species, such as mice [120].

Finally, in mouse oocytes, the transport of water and glycerol through the plasma membrane occurs via simple diffusion. While membrane channels are involved in the transit of water and cryoprotectants [121,122], the fact that mouse oocytes show low permeability to water and glycerol suggests that AQPs are not very abundant in these cells [123]. In spite of this, the treatment of mouse oocytes with mercury chloride, which inhibits all AQPs except AQP7, partially blocks water flow through the plasma membrane [124,125], thus suggesting that not only AQP7 but also other mercury-sensitive AQPs play an important function in water transport in mouse oocytes. This, together with the partial (but not complete) blocking of water flow through the plasma membrane with mercury chloride [124], suggests that not only AQP7 is involved in the osmoregulation of mouse oocytes.

### 4.2. Aquaporins, Oocyte Functionality and Female Infertility

Considering that, in mature oocytes, water seems to cross the plasma membrane mainly through simple diffusion in the different species of mammals that have been studied thus far, AQPs seem to have a less relevant role in this cell type than in sperm. Different studies, however, support the idea that AQPs are crucial during oocyte maturation. This process requires an interaction between granulosa cells and oocytes, and AQPs have been proposed to be involved in these interactions.

Jo et al. [126] described an increase in *Aqp3* mRNA expression during in vivo and in vitro oocyte maturation, which decreased in mature oocytes. Moreover, controlled ovarian hyperstimulation in mouse is known to reduce *Aqp3* mRNA and AQP3 protein expression in oocytes [121] Furthermore, water permeability and swelling in response to osmotic stress in oocytes from female mice that have undergone controlled ovarian hyperstimulation are impaired, as, too, are fertilization rates, which confirms the importance of AQP3 for the quality of these oocytes [121].

In rat oocytes, water permeability is higher in oocytes collected in the proestrus than in those collected in the estrus. In addition, while water permeability in oocytes collected during the proestrus is blocked by mercury, this parameter is not sensitive to this agent during the estrus [57]. Together with the changes in *Aqp9* expression between these two phases, these data suggest that AQP9 plays a relevant function in proestrus oocytes in the rat.

It is important to highlight that, in a similar fashion to that reported for sperm, the redox equilibrium is of high importance for oocyte integrity, since oxidative stress can damage oocyte structures which, in turn, may compromise their function [122]. Nevertheless, and to the best of our knowledge, no study has investigated the implication of AQPs in the transport of H_2_O_2_, either in mitochondria or through the plasma membrane, in mammalian oocytes. Considering the relevance of oxidative stress in oocytes for female fertility, further research addressing the role that AQPs exert in oocyte physiology is highly warranted.

### 4.3. Aquaporins and Oocyte Cryopreservation

The importance of oocyte cryopreservation has increased recently because of the need for a standard approach that preserves fertility in women suffering from diseases requiring treatments with detrimental effects on germ cells or with high risk of premature ovarian insufficiency (reviewed in [123]). In addition, fertility postponing for social reasons has led many women to preserve oocytes in view of the decline in quality with age due to expectations of lacking a stable partner, career choices or financial issues. Cryopreservation is also useful for long-term storage in human oocyte banks, which makes the process easier by eliminating the need for cycle synchronization between donors and receptors (reviewed in [123]). At present, vitrification is considered to be more efficient than slow-freezing when cryopreserving mammalian oocytes [124]. Vitrification allows the rapid solidification of cells and the extracellular medium, in which both are converted into a glass-like state that avoids the formation of ice [125]. During vitrification, rapid cooling takes place in the presence of high concentrations of permeable and non-permeable cryoprotectants. Permeability to water and cryoprotectants is thus of high importance to minimize the time of exposure to high concentrations of cryoprotectants at temperatures of 25 °C or higher, at which these molecules are toxic for oocytes. Increase of cooling/warming rates and water dehydration to avoid intracellular crystal formation are crucial to boosting the resilience of oocytes to vitrification [125,127]. Since the cryoprotectants preferably used over the last decades are propylene glycol, ethylene glycol, DMSO and sucrose, and glycerol is not routinely employed [125]. the main role of AQPs during oocyte vitrification must be strictly related to their water permeability. 

Permeability to glycerol, which could be considered a potential alternative cryoprotectant for vitrification, has also been studied in different species. It has been previously stated that, in mouse oocytes, the transport of water and glycerol through the plasma membrane occurs mainly via simple diffusion, which suggests that AQPs are not very abundant in these cells [128]. In spite of this, oocytes from an *Aqp7* knockout mouse model present reduced survival after vitrification [129]. This, together with the partial (but not complete) blocking of water flow through the plasma membrane with mercury chloride [130], suggests that AQP7 is not only involved in osmoregulation but also in the cryotolerance of mouse oocytes. In fact, the hyperosmotic stress caused by the presence of different cryoprotectants in the medium, including DMSO, ethylene glycol and sucrose, increases the expression of the AQP7 protein in the plasma membrane, but not that of AQP3 and AQP9 [129,130]. The molecule with the highest impact on AQP7 expression is DMSO, and its presence in the vitrification medium is associated with a higher water flow through the plasma membrane, which allows the oocyte to achieve the osmotic balance within a reduced time compared to other cryoprotectants [129,130]. Regarding the mechanism through which this increase in protein expression occurs, Tan et al. [129] proposed that hyperosmotic stress could trigger signaling pathways involving PI3K and PKC that would activate Aurora A (Figure 6). Aurora A would then phosphorylate cytoplasmic polyadenylation element-binding protein (CPEB), which would be released from already existing mRNAs, thereby allowing their translation. This mechanism of mRNA translational control by CPEB was previously described in the oocyte as crucial for many developmental processes [131]. In the specific case of AQPs, F-actin has been suggested to carry out the translocation of AQP7 from the cytoplasm to the plasma membrane under hyperosmotic stress, allowing AQP7 to facilitate the transport of water and other molecules, such as cryoprotectants [129]. 

With regard to other AQPs, oocytes from an *Aqp3* knockout mouse model present lower resilience to vitrification, although the relevance of this protein in osmoregulation during cryopreservation is substantially more modest than that of AQP7 [129]. The exogenous expression of AQP3 in mouse oocytes, nevertheless, improves their survival of cryopreservation and does not alter their capacity to be fertilized [132,133,134]. 

In human oocytes, cryopreservation causes a decrease in the mRNA levels of *AQP1*, *AQP2* and *AQP11*, as well as in the expression of AQP1 protein in the plasma membrane [58]. The presence of melatonin during the cryopreservation process, however, avoids the decrease in the mRNA and protein levels of these AQPs. This occurs in conjunction with a better oocyte cryotolerance, including the maintenance of morphology, ultrastructure, mitochondrial function and water permeability, and higher fertilization rates, embryo quality and blastocyst rates [58].

## 5. Conclusions

Aquaporins (AQPs) are ubiquitously expressed in different species, organs, tissues and cells and they are crucial for the transport of water and small solutes across cell membranes. Focusing on mammalian gametes, AQP1, AQP3, AQP7, AQP8, AQP9 and AQP11 have been identified in sperm, whereas AQP1, AQP2, AQP3, AQP7, AQP9 and AQP11 have been reported to be present in oocytes. In both cell types, AQP3 and AQP7 have been thoroughly studied in terms of physiological relevance, and mounting evidence supports the proposition that AQP3 is crucial for the function of both gametes. It must be highlighted that most of these AQPs belong to the group of GLPs which, in addition to water, are permeable to other solutes, including glycerol and H_2_O_2_. It has been suggested that AQP7 may be involved in the influx of glycerol, which would be used as an energy source by sperm, but further research is needed to establish this relationship. In addition, both water and glycerol permeability are highly relevant during cryopreservation because quick efflux of water and influx of cryoprotectants are crucial to reducing the formation of intracellular crystals and shortening cooling time, and the rapid rehydration and outflow of cryoprotectants are vital to avoiding toxic effects on gametes. Specifically in sperm, AQPs are also relevant to overcome the major osmotic stress that they undergo immediately after ejaculation, when they enter the female reproductive tract. This osmoadaptation is required for motility activation. Future research should aim to define the exact sets of AQPs that are present in the oocytes from different species, as the present studies envisage certain AQPs and their roles but do not provide complete information on the entire set of AQPs. This is of high importance because, in sperm, AQPs are known to compensate the role of non-functional members. Since AQPs are present in all species and cells, and appear to be important for gamete cryotolerance, further research aimed at identifying their localization and addressing their involvement in cryotolerance is warranted. Related to this, not only can humans and farm animals benefit from this long-term preservation strategy but also endangered species, the creation of biobanks being of great importance for biodiversity and conservation.

## Figures and Tables

**Figure 1 animals-12-00573-f001:**
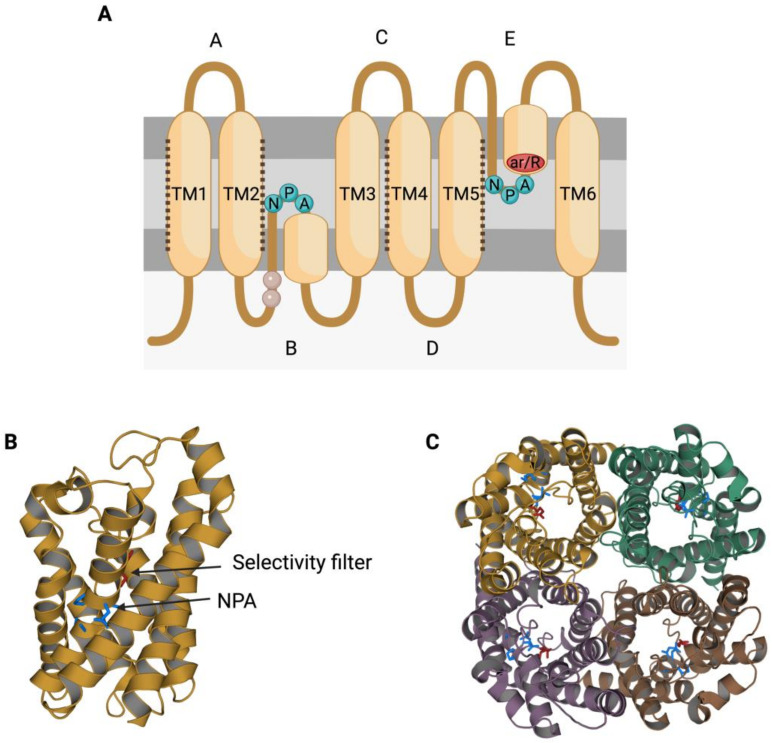
Structural characteristics of the family of aquaporins (AQPs). (**A**) Aquaporins present six transmembrane α-helices (TM1-6) that are connected through loops (A–E). Loops B and E are half-transmembrane helices oriented towards the center of the pore and present an NPA (asparagine, proline, alanine) motif that is highly conserved. In loop E, there is also a highly conserved arginine (R). Some residues from loop B have been suggested to be involved in AQPs’ mechanosensitivity (light residues). Transmembrane helices TM1, TM2, TM4 and TM5 interact with the TM of the adjacent monomers from an AQP tetramer (dark dot lines). (**B**) Each monomer folds in an hour-glass conformation, and the region near the R from loop E is formed by aromatic residues. This region is known as the aromatic/arginine (ar/R) region but also as a selectivity filter since it forms the narrowest point of the AQP pore. After folding, the two NPA motifs are in the same region, which is also highly conserved. (**C**) The quaternary structure of AQPs consists of the formation of tetramers, where each monomer has its own functional pore. This figure is based on PDB structure 6QZJ from AQP7.

**Figure 3 animals-12-00573-f003:**
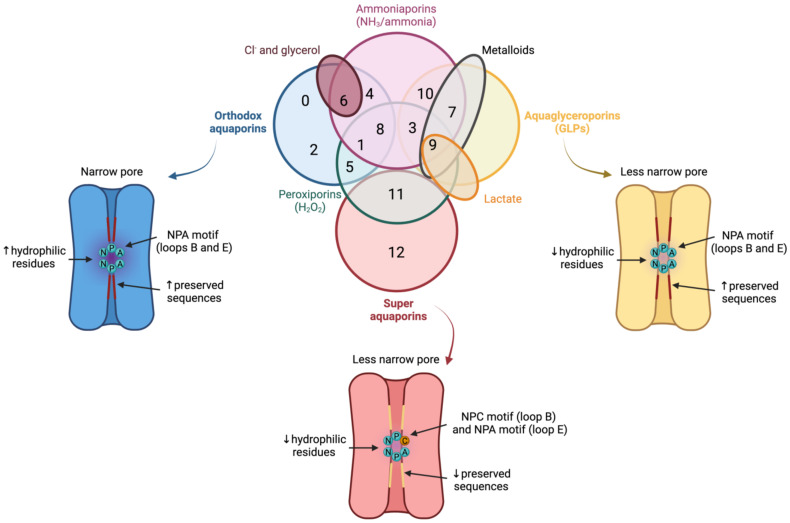
Classification of aquaporins (AQPs) based on their homology and specific permeability to different molecules. The three main groups into which AQPs are classified are: orthodox AQPs (blue), aquaglyceroporins (GLPs; yellow) and superaquaporins (superAQPs; red). The structural characteristics of these three groups are graphically represented. In addition to the classical classification, some AQPs present permeability to H_2_O_2_ and they are considered peroxiporins (green). Similarly, some members of this family of proteins are considered ammoniaporins (pink) since they are permeable to ammonia and/or to NH_3_. Finally, certain members present permeability to other molecules, such as Cl^−^ (dark red), metalloids (grey) or lactate (orange). NPA motif (asparagine, proline, alanine); NPC motif (asparagine, proline, cysteine). Image based on [21,22,23].

**Figure 5 animals-12-00573-f005:**
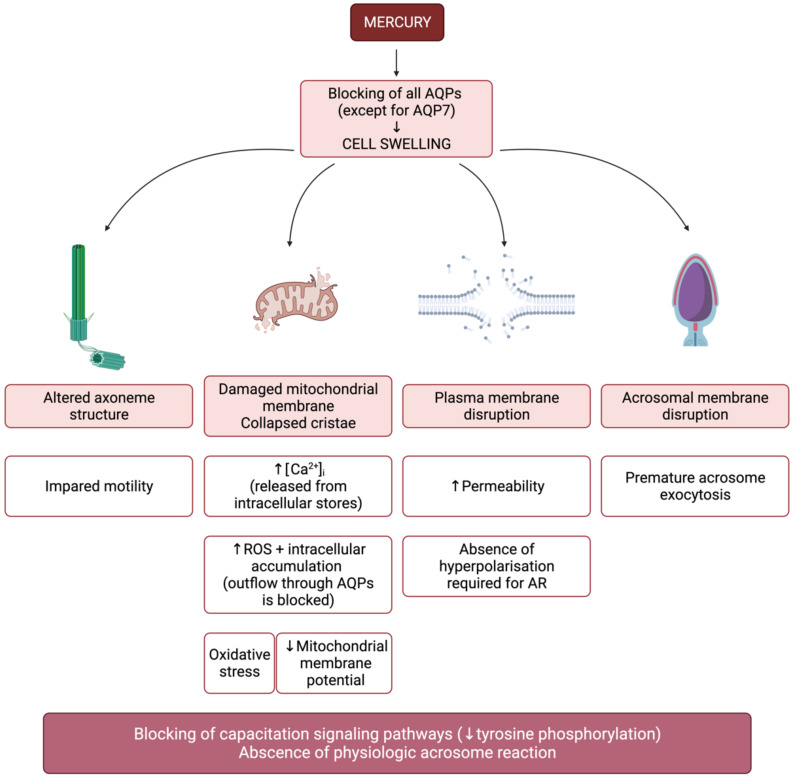
Blocking of aquaporins (AQPs) through mercury causes cell swelling. From a structural point of view, the plasma membrane is disrupted due to stretching and mitochondrial and acrosomal membranes swell. Axoneme structure is also altered, which underlies sperm motility impairment. Mitochondria show collapsed cristae, which decreases energy production. In addition, mitochondrial membrane disruption causes a decrease in mitochondrial membrane potential, release of Ca^2+^ and higher levels of reactive oxygen species (ROS). Since AQPs are permeable to H_2_O_2_, their blockade avoids detoxification and ROS accumulate intracellularly. As previously stated, small amounts of ROS are needed for capacitation, but high concentrations of these molecules have an inhibitory effect on capacitation signaling pathways, which can be assessed through the levels of tyrosine phosphorylation. On the other hand, plasma membrane disruption increases its permeability, which reduces its hyperpolarization, an essential event to trigger the acrosome reaction. Finally, disruption of the acrosomal membrane triggers premature exocytosis and thus sperm cannot undergo a physiological acrosome reaction.

**Figure 6 animals-12-00573-f006:**
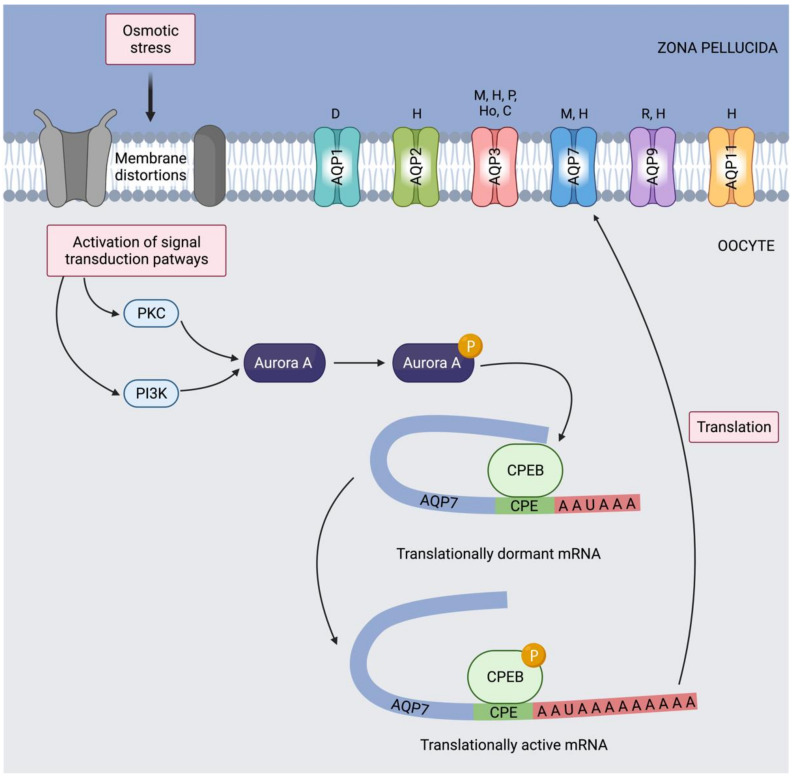
Potential mechanism of translational regulation of aquaporins (AQPs) in mammalian oocytes. Mechanosensitive transmembrane proteins, in response to osmotic stress, trigger signaling pathways that involve protein kinase C (PKC) and phosphoinositide 3-kinase (PI3K). These kinases activate Aurora A, which phosphorylates cytoplasmic polyadenylation element-binding protein (CPEB) from translationally inactive (dormant) mRNAs. These mRNAs then become translationally active, and the resulting AQP7 protein is translocated to the plasma membrane. It has not yet been elucidated whether this mechanism is also involved in the translational regulation of the other AQPs identified in mammalian oocytes (P, pig; Ho, horse; C, cattle; D, dog; H, human; M, mouse; R, rat).

## Data Availability

Not applicable.

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
