# Peer review of "Relevance of Aquaporins for Gamete Function and Cryopreservation"

_animals, 2022, doi:10.3390/ani12050573_

Round 1
Reviewer 1 Report
Dear Marc
Congratulations on this review on the role of Aquaporins in the regulation of gamete function. I have critically reviewed the manuscript and really found it to be novel in its composition and information for the scientific fraternity. It is still fascinating to understand the gamete function for successful fertilization and embryo development. I am working in this area of sperm signaling and communication system and well understand there is a big gap to understand before fully understanding this process. The manuscript is well written and novel in information. The review provides new insights into the mechanisms of Aquaporin function in the regulation of game function. Even I found this manuscript suitable for publication; still I feel there is a number of a suggestion that may be addressed by you before this manuscript will finally be suitable for acceptance. My suggestions for this manuscript are appended below.
- The present review attempts to provide novel insights into the functional relevance of Aquaporins in spermatozoa during their journey and cryopreservation. These are two different states where the spermatozoa encounter an ever-changing extracellular milieu. The authors have touched on one of the most significant parts of sperm's function through its membrane by the regulation of water flow across its membrane. I suggest the authors here provide a flow diagram that will represent the role of Aquaporins in the regulation of sperm function.
- I suggest the authors draw figures from epididymis to Oviduct along with different functional areas and try to show how these Aquaporins are associated with the regulation of sperm function.
- A brief note on the possibility of evolution of these Aquaporins in spermatozoa may be outlined looking at the functional events of spermatozoa during its life cycle starting from the epididymis to Oviduct.
- The authors claim that Aquaporins regulate water fluxing mechanisms along with the regulation of the influx of glycerol. This is one of the most common cryoprotectants used in semen freezing. I suggest the authors provide brief insights into the action of glycerol with Aquaporins and if possible through a figure or chart.
- The middle part of the review is quite interesting where the authors are showing the interplay between the Aquaporins, HgCl2, and ROS in the regulation of sperm function. There is a recent report published depicts that motility reduction by mercuric chloride-is due to mitochondrial damage. Mitochondrial cristae show damage in goat spermatozoa treated with HgCl2 (Kushwaha et al., 2021, Scientific Reports). It is well known that mercuric chloride affects the Aquaporins, and it is the inhibition of aquaporins along with cryopreservation-associated oxidative damage that results in mitochondrial damage and motility inhibition. Hence, I suggest the authors put a figure showing the possible mechanistic pathways of Aquaporins in the regulation of spermatozoa function along with HgCl2 as you have mentioned these mechanisms in your manuscript.
- Are Aquaporins are associated with cryotolerance of spermatozoa during freezing-thawing: Please comment on this with suitable literature in the manuscript.
- There are inter ejaculate, Intra ejaculate, and bull ejaculate variations in cryotolerance and therefore, there is a variation in the post-thaw quality of spermatozoa. If possible and any literature is available, please incorporate it in this manuscript.
- Is there any interplay between Hv1 channels and Aquaporins in the regulation of sperm cryotolerance during freezing-thawing: try to comment on this in your manuscript if any literature is available? Is there any relationship between high freezable semen and low freezable semen in relationship with Aquaporins as studies have shown better cryotolerance is shown by the semen of high freezable semen compared to low freezable semen?
- I suggest incorporating a summary figure showing the various interplay mechanisms, the aquaporins play in the regulation of spermatozoa functional dynamics, e.g. membrane integrity, acrosome integrity, and acrosome reaction.
- How the Aquaporins are responsible for sperm hyperactivity in the oviduct? I am missing this part. Is there any relationship between calcium regulatory pathways and Aquaporin? Having said this, it is also possible that, these may be hypothetical and there is no functional relevance of these in the regulation of sperm capacitation.
These are some of the suggestions to be incorporated in this manuscript and they should be supported by literature if any.
With best wishes
Author Response
Dear Marc
Congratulations on this review on the role of Aquaporins in the regulation of gamete function. I have critically reviewed the manuscript and really found it to be novel in its composition and information for the scientific fraternity. It is still fascinating to understand the gamete function for successful fertilization and embryo development. I am working in this area of sperm signaling and communication system and well understand there is a big gap to understand before fully understanding this process. The manuscript is well written and novel in information. The review provides new insights into the mechanisms of Aquaporin function in the regulation of game function. Even I found this manuscript suitable for publication; still I feel there is a number of a suggestion that may be addressed by you before this manuscript will finally be suitable for acceptance. My suggestions for this manuscript are appended below.
Comment: The present review attempts to provide novel insights into the functional relevance of Aquaporins in spermatozoa during their journey and cryopreservation. These are two different states where the spermatozoa encounter an ever-changing extracellular milieu. The authors have touched on one of the most significant parts of sperm's function through its membrane by the regulation of water flow across its membrane. I suggest the authors here provide a flow diagram that will represent the role of Aquaporins in the regulation of sperm function.
Answer: Thank you for your suggestion. Figure 4 has been added to the manuscript. It contains a schematic representation of the functions that AQPs exert and the intracellular mechanisms in which they might be involved.
Comment: I suggest the authors draw figures from epididymis to Oviduct along with different functional areas and try to show how these Aquaporins are associated with the regulation of sperm function.
Answer: Thank you, we appreciate your comment. Even though we agree that a figure that contains this information would be very interesting, we advise that this review is not focused on this aspect of the role of AQPs. If we rightly understand your comment, the figure that you suggested adding would consist of specifying which AQPs are present in the male and female reproductive tracts. Since this review focuses on the role of AQPs in mature gametes, we believe that this figure would step out a bit from the main subject of the work.
Comment: A brief note on the possibility of evolution of these Aquaporins in spermatozoa may be outlined looking at the functional events of spermatozoa during its life cycle starting from the epididymis to Oviduct.
Answer: Thank you for your comment. We are not sure about what the reviewer meant and what they would refer at. Could they please explain their suggestion?
Comment: The authors claim that Aquaporins regulate water fluxing mechanisms along with the regulation of the influx of glycerol. This is one of the most common cryoprotectants used in semen freezing. I suggest the authors provide brief insights into the action of glycerol with Aquaporins and if possible through a figure or chart.
Answer: Thank you for your suggestion. The relevance of glycerol for sperm and how AQPs might be involved has been detailed in sections 2.2, 2.1.1 and 2.2.2.
Comment: The middle part of the review is quite interesting where the authors are showing the interplay between the Aquaporins, HgCl2, and ROS in the regulation of sperm function. There is a recent report published depicts that motility reduction by mercuric chloride-is due to mitochondrial damage. Mitochondrial cristae show damage in goat spermatozoa treated with HgCl2 (Kushawaha et al., 2021, Scientific Reports). It is well known that mercuric chloride affects the Aquaporins, and it is the inhibition of aquaporins along with cryopreservation-associated oxidative damage that results in mitochondrial damage and motility inhibition. Hence, I suggest the authors put a figure showing the possible mechanistic pathways of Aquaporins in the regulation of spermatozoa function along with HgCl2 as you have mentioned these mechanisms in your manuscript.
Answer: Thank you very much for your valuable suggestion. A figure showing the potential mechanistic pathways through which AQPs can modulate the sperm response (including alterations) to the presence of mercury has been added (Figure 5). In addition, a brief discussion about this matter has been added to section 3.2.3.
Comment: Are Aquaporins are associated with cryotolerance of spermatozoa during freezing-thawing: Please comment on this with suitable literature in the manuscript.
Answer: The relationship between Aquaporins and cryotolerance, based on the literature available thus far, is explained in lines 349-374.
Comment: There are inter ejaculate, Intra ejaculate, and bull ejaculate variations in cryotolerance and therefore, there is a variation in the post-thaw quality of spermatozoa. If possible and any literature is available, please incorporate it in this manuscript.
Answer: Thank you for your suggestion. References for stallion, bull and boar good and poor freezability ejacualtes have been added (section 3.3).
Comment: Is there any interplay between Hv1 channels and Aquaporins in the regulation of sperm cryotolerance during freezing-thawing: try to comment on this in your manuscript if any literature is available?
Answer: Both AQPs and Hv1 channel are relevant during sperm cryopreservation (Delgado-Bermúdez et al. Int. J. Mol. Sci. 2021, 22, 1646). Nevertheless, to the best of our knowledge, there is no literature available on the potential interplay between these two types of channels during sperm cryopreservation. Both channels are essential to reduce oxidative stress. On the one hand, AQPs blocking ends up causing cell swelling and mitochondrial damage, which leads to an increase in ROS levels and alterations in plasma membrane. On the other hand, when Hv1 is blocked, intracellular acidification caused by the accumulation of H+ also favors lipid peroxidation through the formation of hydroxyl radicals, thus leading to plasma membrane disorder.
In spite of all the aforementioned, however, we are not completely sure of referring to this possibility, because, to the best of our knowledge, a direct interaction or a coordinating mechanism has not been described for this pair of channels. In the case, therefore, we could risk of speculating too much about the potential link. All that being said, if the reviewer disagrees (and with the consent of the Editor), we can have a look through the issue again and include more information in the Manuscript body.
Comment: Is there any relationship between high freezable semen and low freezable semen in relationship with Aquaporins as studies have shown better cryotolerance is shown by the semen of high freezable semen compared to low freezable semen?
Answer: The relationship between Aquaporins and cryotolerance is explained in lines 349-374. To sum up, high freezability ejaculates present higher levels of certain AQPs in different species, and in addition, the inhibition of certain AQPs has a negative effect on cryotolerance.
Comment: I suggest incorporating a summary figure showing the various interplay mechanisms, the aquaporins play in the regulation of spermatozoa functional dynamics, e.g. membrane integrity, acrosome integrity, and acrosome reaction.
Answer: Thank you for your comment. Figure 4 summarizes these concepts.
Comment: How the Aquaporins are responsible for sperm hyperactivity in the oviduct? I am missing this part. Is there any relationship between calcium regulatory pathways and Aquaporin? Having said this, it is also possible that, these may be hypothetical and there is no functional relevance of these in the regulation of sperm capacitation.
Answer: Thank you very much for this comment. Following the reviewer’s request, the first paragraph of section 3.2 has been modified, since, to the best of our knowledge, no study confirming the direct relationship between calcium channels and AQPs has been published.
“The relationship between AQPs and sperm function has been evaluated in previous studies. Essentially, the sperm capacity to regulate volume is crucial for fertility, since cell swelling in response to osmotic stress causes tail bending, which can, in turn, affect sperm motility (reviewed in [68]). Nevertheless, osmotic changes in response to the hypotonic shock that sperm undergo when they enter the female reproductive tract are essential to elicit sperm capacitation. On the one hand, changes in sperm volume in response to the osmotic shock trigger the opening of mechanosensitive calcium channels and thus, elicit calcium influx, which is one of the first events that occurs during capacitation [69]. On the other hand, acrosomal swelling is required for a physiological acrosome reaction [70]. In addition, strict regulation of H2O2 concentration is essential for the activation of hyperactivation of motility and the acrosome reaction during capacitation, despite the mechanism is not yet under-stood [71]. Since some members of this family are peroxiporins, their potential involvement in acrosome reaction can be suggested. The role of AQPs in fertility is, therefore, evidenced by their involvement in both motility and capacitation-associated events, which are crucial for sperm to succeed in fertilization.”
These are some of the suggestions to be incorporated in this manuscript and they should be supported by literature if any.
With best wishes
Answer: We appreciate the time and effort of the reviewer, while trying to improve the quality of this review article. We do think that all their suggestions have much improved the quality of the paper.
Reviewer 2 Report
The authors have published a similar review Yeste et al. in Reprod Dom Anim. 2017;52(Suppl. 4):12–27.; however presented review is enriched with new data and structured differently. The authors quote 17 self-citations, which are adequate given the content of the review.
I have suggestions mainly to change the structure of Chapter 4 and to add a diagram and scheme for a better understanding of the issue.
Simply Summary should, in my opinion, contain information for the general public and, conversely, the Abstract should be more professional, so I would exchange some information between these paragraphs. Moreover, the authors should shorten the Simply Summary.
I do not understand the sentence (l. 20-23): Because not only is cryopreservation of high importance for fertility... Please, simplify and adjust this.
I suggest removing the term aquaporins from the Keywords and adding other terms.
Figure 1 should be below the text (l. 79-105)
Please, mark in the picture aromatic / arginine (ar / R)
Please, always specify which image you are referring to exactly in the text, Figure 1A or 1B?
I suggest to put in the diagram the classification of AQPs into 3 groups, in the diagram there would be a picture showing the different structure of AQPs in groups and how the individual groups are significant.
I assume the species in Table 1 are listed chronologically by study year. I suggest to write this fact in the description of the table or in the text where there is a reference to the table.
I would include an introductory paragraph on the summary for the detection of AQPs in sperm and oocytes, followed by a table.
The athors should be added a scheme where the AQPs were located in the sperm and in what species.
If the authors state Aquaporin at the beginning of the sentence, then enter its abbreviation (AQP3) in parentheses (eg 173, 175, 178, 179, 183, etc.)
Subchapters to chapters AQPs in sperm and oocyte are not equivalent. Please, try to structure better - localization of AQPs in sperm / oocyte, physiological function of AQPs in sperm / oocyte, role of AQPs in sperm / oocyte during cryopreservation.
I do not understand the sentence (l. 20-23): Because not only is cryopreservation of high importance for fertility... Please, simplify and adjust this.
I suggest removing the term aquaporins from the Keywords and adding other terms.
Figure 1 should be below the text (l. 79-105)
Please, mark in the picture aromatic / arginine (ar / R)
Please, always specify which image you are referring to exactly in the text, Figure 1A or 1B?
I suggest to put in the diagram the classification of AQPs into 3 groups, in the diagram there would be a picture showing the different structure of AQPs in groups and how the individual groups are significant.
I assume the species in Table 1 are listed chronologically by study year. I suggest to write this fact in the description of the table or in the text where there is a reference to the table.
I would include an introductory paragraph on the summary for the detection of AQPs in sperm and oocytes, followed by a table.
The athors should be added a scheme where the AQPs were located in the sperm and in what species.
If the authors state Aquaporin at the beginning of the sentence, then enter its abbreviation (AQP3) in parentheses (eg 173, 175, 178, 179, 183, etc.)
Subchapters to chapters AQPs in sperm and oocyte are not equivalent. Please, try to structure better - localization of AQPs in sperm / oocyte, physiological function of AQPs in sperm / oocyte, role of AQPs in sperm / oocyte during cryopreservation.
Author Response
The authors have published a similar review Yeste et al. in Reprod Dom Anim. 2017;52(Suppl. 4):12–27.; however presented review is enriched with new data and structured differently. The authors quote 17 self-citations, which are adequate given the content of the review. I have suggestions mainly to change the structure of Chapter 4 and to add a diagram and scheme for a better understanding of the issue.
Comment: Simply Summary should, in my opinion, contain information for the general public and, conversely, the Abstract should be more professional, so I would exchange some information between these paragraphs. Moreover, the authors should shorten the Simply Summary.
Answer: Thank you for your comment. Both the Simple summary and the Abstract have been modified following the reviewer’s suggestion.
Comment: I do not understand the sentence (l. 20-23): Because not only is cryopreservation of high importance for fertility... Please, simplify and adjust this.
Answer: Thank you for your comment. The sentence has been revised and its meaning clarified.
Comment: I suggest removing the term aquaporins from the Keywords and adding other terms.
Answer: Thank you for your suggestion. We have modified the keywords accordingly: mammals; oocyte; sperm; water channels; physiology; cryopreservation.
Comment: Figure 1 should be below the text (l. 79-105)
Answer: This figure has been relocated accordingly. Thank you for your suggestion.
Comment: Please, mark in the picture aromatic / arginine (ar / R)
Answer: The figure has been modified accordingly.
Comment: Please, always specify which image you are referring to exactly in the text, Figure 1A or 1B?
Answer: Thank you for your suggestion. References to Figure1 in the text have been revised and modified accordingly.
Comment: I suggest to put in the diagram the classification of AQPs into 3 groups, in the diagram there would be a picture showing the different structure of AQPs in groups and how the individual groups are significant.
Answer: Thank you for your comment. Figure 3 has been added to the manuscript.
Comment: I assume the species in Table 1 are listed chronologically by study year. I suggest to write this fact in the description of the table or in the text where there is a reference to the table.
Answer: Thank you for your comment. The table has been revised to check that all species are listed chronologically by the year of study, and this has been referred to where the Table is mentioned for the first time: “A summary of the members of this family that have been explored in gametes to date is presented in Table 1, which includes the different species in which they have been studied in a chronological order.”
Comment: I would include an introductory paragraph on the summary for the detection of AQPs in sperm and oocytes, followed by a table.
Answer: Thank you for your suggestion. In sections 3 and 4, where the table is cited in the text, more details on the detection of AQPs in sperm and oocytes has been added. In addition, the methodology through which AQPs were detected has been added to the table.
Comment: The athors should be added a scheme where the AQPs were located in the sperm and in what species.
Answer: Thank you for your suggestion. Figure 4 has been added to the manuscript and additional specifications have been added to Section 3.
Comment: If the authors state Aquaporin at the beginning of the sentence, then enter its abbreviation (AQP3) in parentheses (eg 173, 175, 178, 179, 183, etc.)
Answer: Thank you for your suggestion. We have made this change along the manuscript.
Comment: Subchapters to chapters AQPs in sperm and oocyte are not equivalent. Please, try to structure better - localization of AQPs in sperm / oocyte, physiological function of AQPs in sperm / oocyte, role of AQPs in sperm / oocyte during cryopreservation.
Answer: Thank you for your comment. The information in the chapter “Aquaporins in mammalian oocytes” has been reorganized following the reviewer’s request.
Reviewer 3 Report
The role of aquaporins on gamete functionality and their potential implications in cryopreservation is described clearly and with good writing. Compared to a previous review on the same topic by the same author, new information and attention to the female gamete have been introduced.
I, therefore, have no major criticisms to make, except that in addressing the traffic of water and other components through the plasma membrane of gametes, in spite of repeating things already addressed in the previous review, no reference was made to ionic currents which together with water and cryoprotectants are heavily involved in the regulation of cellular osmotic activity.
The discrimination that is repeatedly reiterated between osmotic changes faced by male gametes compared to what happens to female gametes perhaps should be reconsidered, by virtue of the ionic changes that occur during the oocyte maturation as well as during its activation with fertilization (fertilization currents, increased intracytoplasmic calcium levels). Among other things, as the Authors reported on Line 457-460, the alteration of the water permeability and swelling in response to osmotic stress in the oocytes obtained by follicular stimulation involves a reduction in their fertilizing capacity. Furthermore, the higher osmolarity to which spermatozoa undergo during their stationing and epididymal transit, as reported by Cooper, is not limited to the tail of the epididymis (cauda) and is not resolved only in the female tract but also by meeting the other components of the seminal plasma.
Small indications:
Line 20-23. Please, re-write this sentence because unclear.
Line 107. “Proposed…” who did propose this model?
Line 238. Replace "concludes ..." with "suggested that the observed effects were ..."
Line 305. “…from other species of mammals” in truth, there are only reference to the pig.
Line 338. Glycerol is not the most used CPA for almost all species (again on line 366)
Line 477. The expression of exogenous (human and zebrafish) AQP3
Author Response
The role of aquaporins on gamete functionality and their potential implications in cryopreservation is described clearly and with good writing. Compared to a previous review on the same topic by the same author, new information and attention to the female gamete have been introduced.
Comment: I, therefore, have no major criticisms to make, except that in addressing the traffic of water and other components through the plasma membrane of gametes, in spite of repeating things already addressed in the previous review, no reference was made to ionic currents which together with water and cryoprotectants are heavily involved in the regulation of cellular osmotic activity.
Answer: Thank you for your suggestion. This concept was explained in sections 2 and 3.1, and is represented in Figure 2. An additional description of the process has been added to the main text, which now includes the expression “ion currents”, since it had only been described as “regulatory volume decrease (RVD)”.
Comment: The discrimination that is repeatedly reiterated between osmotic changes faced by male gametes compared to what happens to female gametes perhaps should be reconsidered, by virtue of the ionic changes that occur during the oocyte maturation as well as during its activation with fertilization (fertilization currents, increased intracytoplasmic calcium levels). Among other things, as the Authors reported on Line 457-460, the alteration of the water permeability and swelling in response to osmotic stress in the oocytes obtained by follicular stimulation involves a reduction in their fertilizing capacity.
Answer: Thank you very much for your comment. We revised Section 3.2 following the reviewer’s request, and these two points have been included.
Comment: Furthermore, the higher osmolarity to which spermatozoa undergo during their stationing and epididymal transit, as reported by Cooper, is not limited to the tail of the epididymis (cauda) and is not resolved only in the female tract but also by meeting the other components of the seminal plasma.
Answer: Thank you for your comment. A more extended explanation has been added in Section 3.1.: “In fact, during their transit through the epididymis, sperm acquire their osmoadaptability, since the extracellular medium is progressively more hyperosmotic from the caput to the cauda. With this purpose, sperm uptake osmolytes from the epidydimal plasma, which allows the counteraction of the lower osmolality at the female reproductive tract after ejaculation (Yeung, et al. Mol. Cell. Endocrinol. 2006, 250, 98–105.). One must also consider that, at the time of ejaculation, sperm encounter the seminal plasma. There are differences between different species of mammals in terms of differential osmolality between cauda epidydimis, seminal plasma and the oviduct. On the one hand, in the bovine, the epididymis, uterine, and oviductal environments are hyperosmotic compared to the isoosmotic seminal plasma. In humans, mice, and rats, osmolality is progressively lower from the epididymis to the seminal plasma (after ejaculation) and then to the uterus; whereas in ruminants, the uterus and the epididymis are relatively hyperosmotic with regard to the seminal plasma (Lavanya, et al. Andrology 2022, 10, 92–104).”
Small indications:
Comment: Line 20-23. Please, re-write this sentence because unclear.
Answer: Thank you for your comment. This sentence has been rewritten accordingly.
Comment: Line 107. “Proposed…” who did propose this model?
Answer: Thank you for your comment. Figure 2 capture has been modified accordingly.
Comment: Line 238. Replace "concludes ..." with "suggested that the observed effects were ..."
Answer: Thank you for your suggestion. The sentence has been modified accordingly.
Comment: Line 305. “…from other species of mammals” in truth, there are only reference to the pig.
Answer: Thank you for your suggestion. After the revision of the manuscript, additional data have been added to this subsection and, now, it also contains information regarding goat sperm. Therefore, we consider that the title could remain unchanged.
Comment: Line 338. Glycerol is not the most used CPA for almost all species (again on line 366)
Answer: Thank you for your suggestion. We have modified this sentence indicating that glycerol is used “in different livestock species”.
Comment: Line 477. The expression of exogenous (human and zebrafish) AQP3
Answer: Thank you for your comment. The sentence has been changed accordingly.
Reviewer 4 Report
Figure 1 is a creation of the authors? otherwise say the source
Author Response
Comment: Figure 1 is a creation of the authors? otherwise say the source
Answer: Thank you for your comment. All figures are a creation of the authors. A sentence in the Acknowledgements section has been added to specify that they were created with Biorender.
Round 2
Reviewer 2 Report
The authors have significantly improved their manuscript, rearranged the chapters and added clear schemes. I have only one comment, which they did not meet. Please, add for better orientation in the text vs pictures in parentheses the acronym for aquaporin (eg. AQP 3): lines 187-195. Table 1 should be listed immediately after the line 186. Thank you.
Author Response
Thank you very much for your comments. All corrections have been made.